# Effect of Massage with Oil Balanced in Essential Fatty Acids on Development and Lipid Parameters in Very Premature Neonates: A Randomized, Controlled Study

**DOI:** 10.3390/children9040463

**Published:** 2022-03-25

**Authors:** Aurélie Garbi, Martine Armand, Any-Alejandra Beltran-Anzola, Catherine Sarté, Véronique Brévaut-Malaty, Barthélémy Tosello, Catherine Gire

**Affiliations:** 1Department of Neonatology, AP-HM, University Hospital Nord, 13015 Marseille, France; aurelie.garbi@ap-hm.fr (A.G.); any-alejandra.beltran-anzola@univ-amu.fr (A.-A.B.-A.); veronique.brevaut@ap-hm.fr (V.B.-M.); catherine.gire@ap-hm.fr (C.G.); 2Aix Marseille Univ, CNRS, CRMBM, Marseille, France; martine.armand@inserm.fr (M.A.); catherine.sarte@inserm.fr (C.S.); 3Aix Marseille Univ, CERESS, Marseille, France; 4Aix Marseille Univ, CNRS, EFS, ADES, Marseille, France

**Keywords:** very preterm, massage with oil, weight gain, lipid parameters, digestive autonomy, polyunsaturated fatty acids, plasmalogens, nervonic acid

## Abstract

Background: Oil massage versus only massage can increase preterm newborn development, especially weight gain, via a supposed percutaneous absorption of oil lipids, but data are contradictory. Aims: Investigating whether massage with a vegetable oil balanced in essential fatty acids improves neonatal weight gain, and digestive autonomy as proxy for neuro-development outcomes. Methods: A prospective monocentric randomized study was conducted in very premature newborns who received massage with oil (isio4 10 mL/kg/day, *n* = 18) versus with no oil (*n* = 18) for five consecutive days (10-min session twice daily) at a corrected gestational age of 34–35 weeks. Anthropometrics and clinical characteristics were recorded. Plasma triglyceride and total cholesterol concentrations were analyzed with an enzymatic kit. The fatty acid composition (weight%, mg/mL) of total plasma lipids and of red blood cell (RBC) membrane was analyzed by gas chromatography. Results: Weight gain velocity at the end of massage period was 12.3 ± 1.4 g/kg/day with oil vs. 9.8 ± 1.4 g/kg/day with no oil (*p* = 0.1). Digestive autonomy, plasma lipid parameters, polyunsaturated fatty acids in plasma total lipids or in RBC were comparable. The no oil group displayed a higher RBC level in nervonic acid at discharge (4.3 ± 0.2 vs. 3.4 ± 0.2%; *p* = 0.025) and in C18:1n-9 plasmalogen species at the end of the massage period and at discharge (0.73 ± 0.06 vs. 0.48 ± 0.06; 0.92 ± 0.06 vs. 0.69 ± 0.06%; *p* < 0.01), two molecules that are involved in neurodevelopment. Conclusions: The use of isio4 oil did not provide additional benefits for the development of very premature newborns, neither changed lipid metabolism nor polyunsaturated fatty acid biological status, which did not corroborate the existence of a percutaneous route for oil lipid absorption. The reason for different levels of nervonic acid and plasmalogen in RBC remains to be explored.

## 1. Introduction

Every year, an estimated 15 million babies (i.e., more than 1 birth on 10) are born preterm (less than 28 weeks to 37 weeks of gestation), with a rate of 5% to 18% across countries, and this number is rising [1,2]. Preterm birth immediate complications are the leading cause of neonatal death (about 1 million each year), which could be prevented with current cost-effective intensive care at hospital. Weight gain during hospitalization is the main criterion for development and leads to early discharge.

The practice of massage is of interest for increasing newborn weight gain [3,4,5,6]. Indeed, several randomized studies have shown significant weight gain, along with a reduced length of hospital stay, in premature infants who received a pressure massage, regardless of the massage technique being used such as touch massage with kinesthetic, acupressure, or meridian massage [4]. The effect exists from two to three sessions per day lasting 10 to 15 min, for 5 to 30 consecutive days.

In addition, the use of a vegetable oil during massage seems to further increase weight gain but this point is still controversial [4,7]. One suggested mechanism to explain a superior weight gain when using massage with oil compared to only massage is the percutaneous absorption of oil lipids, leading to a higher energy supply and nutrients [8,9]. The fatty acid composition of triglycerides in vegetable oils used in studies varies in the polyunsaturated fatty acid (PUFA) content such as linoleic (LA, C18:2n-6) and alpha-linolenic (ALA, C18:3n-3) acids, which are known as essential fatty acids (EFA) [10]. In few randomized controlled trials in very premature neonates, the impact of the fatty acid composition of the massage oil on serum/plasma triglyceride concentration and fatty acid composition were studied, leading to inconclusive data. Indeed, massage oils rich in LA (safflower oil) or in saturated fatty acids (SFA; coconut oil) resulted in a rise in the corresponding fatty acids in the circulating plasma lipids of the premature newborn through a supposed percutaneous absorption route [8], while no change occurred with sunflower oil, which is rich in LA [11]. Nevertheless, it could be hypothesized that the existence of such a cutaneous route, at least during a specific “window”, could allow for an increase in the bioavailability of LA and ALA provided by vegetable oils, leading to an increase in the long-chain PUFA, arachidonic (ARA, C20:4n-6), and docosahexaenoic (DHA, C22:6n-3) acids that are biosynthesized from the two EFA, respectively, thanks to desaturase and elongase enzymes [12]. Such phenomenon would be of health interest for premature newborns since they have a high need in EFA, ARA, and DHA for brain growth [13,14,15] as well as for general body development [12].

Based on the literature, our study aimed to evaluate the benefits of using a massage oil rich and balanced in LA and ALA on the development of premature infants born before 32 weeks of gestational age, compared to massage with no oil.

Thus, our primary objective was to investigate whether this oil was beneficial for the neonate growth (absolute weight gain, weight gain velocity) and for digestive autonomy as a proxy for neuro-development outcomes.

Our secondary objective was to evaluate the hypothesis of a change in the fatty acid composition of plasma lipid parameters (rise in LA, ALA, and possibly in ARA and DHA) as a result of percutaneous absorption of its triglycerides, as well as a change in PUFA biological status.

## 2. Subjects and Methods

### 2.1. Ethical Approval

The study protocol was approved by the Research Ethics Committee of the Assistance Publique des Hôpitaux de Marseille (AP-HM) with registration numbers No. 11033 from CPP Sud-Méditerranée (authorization obtained on 8 July 2011), No. B110440-80 from AFSSAPS (authorization obtained on 20 July 2011), No. 1233938 from CNIL (methodology of reference MR001 authorized for AP-HM studies on 9 May 2007). Informed consent was signed by the parents or legal representatives. The clinicalTrials.gov identifier is NCT02608151.

### 2.2. Design

This study was monocentric (level 3 maternity unit Hôpital Nord Marseille, France), controlled, open, and randomized into two groups. Recruitment was prospective. The experimental group was represented by very premature neonates receiving oil massage intervention. The control group was represented by very premature neonates receiving massage intervention with no oil. Randomization was established using the OLAN procedure (SAS software) to divide the participants into two parallel and comparable groups.

The primary outcome on which our study was powered was weight gain. To detect a difference in weight gain of 3 g/kg/day (with a pooled SD of 2.7 g/kg/day) [16] at an α-error of 5% and a β-error of 10%, it was estimated that 18 subjects would be required for each study group.

### 2.3. Studied Population

Inclusion criteria were the following: (1) gestational age (GA) between 26 and 31 weeks; (2) neonates with central intravenous perfusion less than 60 mL/kg/day; (3) neonates with more than 48 h weaning from continuous positive airway pressure (CPAP); (4) at least one of the two parents (or legal representatives) benefiting from the French Social Security System; and (5) both parents (or legal representatives) having signed the informed consent.

Exclusion criteria were: (1) neonates who developed a co-morbidity during the massage period, implying an interruption of this practice (e.g., nosocomial infection, necrotizing enterocolitis, respiratory deterioration leading to stop feeding and/or to re-put them on CPAP or more invasive therapy); (2) one of the parents refuses to continue with this study; (3) neonates transferred before accomplishment of all massage sessions; (4) neonates with incomplete massage program; and (5) neonates who passed away.

Non-inclusion criteria: (1) neonates with neurological pathologies (intraventricular hemorrhage grade 3 or 4 as evaluated by brain ultrasound); (2) neonates with infectious disease-causing instability (CRP > 7 mg/mL); and (3) neonates with genetic syndrome, progressive neurologic disease, cleft lip, and palate or other malformations.

### 2.4. Massage Intervention

The massage program was organized for 5 consecutive days (from day 0 to day 5) following two 10-min sessions per day. It was performed at least 30 min after any intervening act and one hour before feeding. Neonates were placed in an environment with temperature set at 25 °C, with reduced light and muted alarms to avoid any stressful stimuli. The same two trained nurses performed the massage and alternatively monitored the neonate, limiting inter-operator variability. Sessions were interrupted in cases of clinical instability, desaturations, and apneas associated with bradycardia. If the neonate did not stabilize in 30 min, the session was postponed.

The massage technique being used by the nurses is as follows. The first points of touch massage are located on the soles of the feet, then is performed several times with touch to slide back to mid-distance of the head of the fibula and the anterior tibia tuberosity and then along the inguinal folds. Then, the fingers press in segments on the sternum (start at the top and go down) and then slip two fingers along the sternum to the lower body. Apply simultaneous pressures on the outer side of the chest, and in the back at the level of D1, D2. From the scapula (hands on both sides), the operator makes a slipped touch down with both hands along the arm to the wrist of the child. A touch of pressure is applied to the wrist for at least three seconds. The massage continues with rotations at the folds of the phalanges. The baby is brought slowly into three quarter lateral, one hand is positioned at the top of the head on the median line and the other hand on the coccyx set distance between coccyx and anus on the median line with simultaneous pressure for at least three seconds. Then, glides are supported from top to bottom on the spine of the neonate with two fingers simultaneously touching. The contact is maintained by placing a hand on the baby, and a rotation is made to bring it back to the dorsal position. The two forefingers are placed and supported in the middle of the eyebrow for at least three seconds, followed by a slide down to the hairline. The thumbs are supported in the temporal region, equidistant from the tail of the eyebrows and the temporal root of the hair, and then gliding in front of the upper edge of the zygomatic arch. To finish, a pressure is exerted and maintained at least three seconds at the level of the flag and lobe of the ear.

A vegetal oil (“isio4”, Lesieur, France) was used at a dose of 10 mL/kg/day as in other studies [8,11]. It is a mix of four different vegetable oils (sunflower oil, high oleic sunflower oil named oleisol, grapeseed oil, and rapeseed oil) for a final composition of 8% saturated fatty acids, 62% of monounsaturated fatty acids, and 30% of PUFA (24.3% LA and 5.7% ALA, ratio LA/ALA: 4.3/1). Bottles were well conditioned by the hospital pharmacy for safety purposes and protected from light to avoid any PUFA damage. The choice of this oil mix was based on its convenience for being rich in EFA with a balanced ratio in LA and ALA and the presence of vitamin E (tocopherol).

The massage program started as soon as the neonates were able to receive enteral feeding via orogastric tube of at least 80 mL/kg/day. Enteral feedings were own mother human milk or donor milk.

### 2.5. Measurements

Anthropometrics (age, weight, length, head circumference) and clinical data were collected at antenatal, at birth, and at postnatal. From the massage intervention period until discharge, the same physician, who was not informed about which group each neonate was randomly assigned to, was in charge of collecting all anthropometrics and clinical data.

Weight was measured daily for each neonate, and the weight gain after massage intervention and between the end of intervention and discharge was calculated as absolute (weight measured at the end of massage period—weight at day 0 of massage; weight at discharge—weight at the end of massage period) and as weight gain velocity (absolute weight gain divided by the weight in kg measured at inclusion or measured at the end of massage period and divided per the total number of days). Head circumference and length were measured once a week until discharge.

### 2.6. Blood Collection and Lipid Analysis

Blood was collected at three periods i.e., at the start of the massage intervention (day 0), the day after the end of the massage period (named day 5 for simplification) and at discharge. Blood samples were drawn (no more than 1 mL/kg) following a specific procedure avoiding pain and at the same time as other needed blood tests, thus avoiding frequent venipuncture.

Blood samples was kept at 4 °C until transfer to the research laboratory where samples were centrifuged for separating plasma, white cells, and red blood cells (RBC). A 0.011 M Tris buffer pH 7.5 was added to the RBC fraction (*v/v*) for lysis. Plasma and RBC were stored at −80 °C until analyses. RBC phospholipid membrane was isolated by centrifugation (30 min, 28,250× *g*, Sorval Centrifuge, 4 °C) of the RBC fraction in Tris-buffer, then removal of the supernatant. The process was repeated until a white pellet was obtained, which was collected with 100 µL of the Tris buffer and stored at −80 °C prior to analysis.

Plasma triglyceride and total cholesterol concentrations were determined by enzymatic colorimetric kits (GPO-PAP and CHOD-PAP, respectively, Roche) on plasma diluted twice.

The fatty acid composition of the plasma total lipids and of the RBC membrane phospholipids as well as the plasmalogen levels of RBC were determined by gas chromatography (GC Clarus 680, PerkinElmer, Waltham, MA, USA; flame ionization detector, Turbochrom software) using a fast fused silica capillary column (BPX 70, 10 m × 0.1 mm i.d., 0.2 µm film thickness, Sigma-Supelco, Bellefonte, PA, USA). A direct methylation step of the samples with acetyl chloride at 100 °C for 1 h, following a procedure previously published [17,18], transformed the fatty acids carried by all lipids into fatty acid methyl esters (FAME) and the specific fatty acids linked by a vinyl-ether bond at the sn-1 position of the glycerol backbone of the plasmalogens into dimethyl acetals (DMA). After an extraction step, the profile of the methylation products (FAME and DMA) was analyzed. A GC oven temperature rise of 20 °C/min was applied from 60 °C to 200 °C, then 7 °C/min to 225 °C with a 1 min hold and finally 160 °C/min to 250 °C with a 1 min hold. Injector was used at 250 °C and detector temperature was set at 280 °C. Carrier gas was hydrogen at a constant pressure of 207 kPa. 

The control of the proportionality between FAME or DMA (C16:0, C18:0, C18:1n-9 and C18:1n-7 species) peak areas (in µV/s), and their identification based on their respective retention time, were performed using an external calibrator FAME standard (GLC 674, Nu-Chek Prep, Waterville, NM, USA) or by using a phospholipid standard rich in plasmalogens and pure C16:0 DMA (purified phosphatidyl ethanolamine from bovine brain, and hexadecanal dimethyl acetal, both from Sigma), respectively [19]. Internal standard (tridecanoic acid, Sigma-Fluka) in a well-known amount was added in the plasma samples prior to the methylation step for individual fatty acid quantitation.

Each FAME and DMA were expressed as weight% of total fatty acids (plasma, RBC) and as mg/mL of plasma. The relative amount of plasmalogens, identified from DMA, was estimated as the percentage ratio between the DMA species found (C16:0, C18:0, C18:1n-9, or C18:1n-7 fatty alcohol) and the corresponding FAME (e.g., [C18:1n-9 DMA/C18:1n-9 ] × 100) [20].

### 2.7. Statistical Analyzes

Data were described by frequencies and percentages or by means and standard deviation (SD). The normality of the continuous variables was estimated using the Shapiro–Wilk test, and Kolmogorov–Smirnov test.

The characteristics of the two groups (oil massage versus no oil massage) were compared by univariate analysis using Chi-2 or Fisher test for categorical variables, and Student *t*-test or Mann–Whitney for continuous variables, in accordance with the variables’ distribution. The threshold for statistical significance was set at a *p*-value < 0.05 for all tests.

The effects of massage on absolute weight gain and weight gain velocity, as well as on the plasma lipid parameters and on fatty acid biological status for the two groups were analyzed using a general linear model for repeated measurement, with two levels by time for weight gain (between day 0 and after 5-day massage/between end of massage and discharge) and with three levels by time for plasma lipid parameters and fatty acid biological status (day 0/day 5/at discharge). A second analysis was performed by including in the model five characteristics selected by their clinical relevance to adjust for any differences between the two groups: birth weight (which correlated with length and head circumference at birth), birth gestational age, corrected gestational age at day 0 of massage intervention, and bronchopulmonary dysplasia status (yes/no).

Enteral autonomy, the duration of hospital stays, and the anthropometric parameters at discharge (age, weight, head circumference and length) were compared between the two groups using the Student *t*-test or Mann–Whitney for continuous variables, in accordance with the variables’ distribution. Linear regression models including the five characteristics selected for their clinical relevance were applied to each variable to adjust for any differences between the two groups. Data were analyzed using the Statistical Package for the Social Sciences (SPSS) version 20.0 software.

## 3. Results

### 3.1. Characteristics of the Two Groups after Randomization

Thirty-six premature neonates born between 1 September 2011 and 1 September 2013 participated in this study (flow chart in Figure 1) and were randomly divided into two groups: the experimental group receiving massage with oil (oil group, *n* = 18) and the control group receiving massage without oil (no oil group, *n* = 18).

All anthropometrics were available for all neonates for the entire study i.e., at antenatal, at birth, at the start (day 0 massage) and at the end of the massage period (day 5 massage), and at discharge.

However, data on digestive autonomy after massage intervention were missing for one neonate in the oil group and four neonates in the no oil group, leading to a final number of 17 neonates and of 14 neonates for digestive autonomy evaluation. In addition, blood samples were missing for three or one neonate(s) at day 5 massage and for two or one neonate(s) at discharge for the oil or the no oil groups, respectively, leading to a final number of 13 or 15 neonates for the lipid parameters and fatty acid status exploration.

Characteristics of the neonates enrolled in the study are described in Table 1.

At antenatal, no difference was observed for the variables considered. At birth, the gestational age, weight, head circumference and length were significantly lower in neonates enrolled in the oil group. Thus, GA and weight at birth were identified as confounding factors to be considered for adjustment for further statistical analysis.

At postnatal, the corrected age (expressed as GA or days) at inclusion (day 0 massage) was moderately higher in the oil group, and more neonates from this group had bronchopulmonary dysplasia (both *p* values close to 0.05). These two parameters were further considered as confounding factors for statistical analysis.

### 3.2. Effect of Oil Massage on Neonate Development

Massage with oil was tolerated as well as massage with no oil as indicated by a comparable tolerance score/20, which was 15.7 ± 2.9 versus 15.7 ± 2.8, respectively. The oil massage did not lead to higher weight (1853 ± 340 g in oil massage group versus 1977 ± 394 g in no oil massage group), absolute weight gain, and weight gain velocity when measured at the end of the massage period compared to no oil massage before or after adjustment to confounders (Table 2). Weight gain velocity ranged from +2.2 to +25.8 g/kg/day in the experimental group and +3.3 to +15.1 g/kg/day with one neonate losing weight (−1.3 g/kg/day) in the control group. At discharge, no difference was observed for weight (2556 ± 439 g in oil massage group versus 2636 ± 591 g in no oil massage group), weight gain, or weight gain velocity from end-to-massage to discharge between the two groups.

Digestive enteral autonomy expressed as corrected GA was not statistically different before and after adjustment on confounders, while the number of days after birth needed for reaching digestive enteral autonomy was significantly lower for the no oil group (*p* = 0.027; Table 3); such difference was erased after adjustment to confounders factors i.e., birth weight and GA, corrected GA at start of massage and bronchopulmonary dysplasia status (*p* = 0.626). No difference was observed for age (except before adjustment to confounders), head circumference, and length at discharge.

The hospital stay calculated from the end of the massage intervention to discharge (number of days at discharge—number of days at end of massage) was not significantly different between the two groups (Table 3).

### 3.3. Effect of Oil Massage on Plasma Lipid Parameters and Fatty Acid Biological Status

Triglyceride and total cholesterol concentrations in plasma were comparable between the two groups at the start and at the end of massage period as well as at discharge (Table 4).

The LA level of total plasma circulating lipids were not different between groups at the three collection times either expressed as % total FA or as mg/mL plasma. The plasma levels of ALA and DHA at day 0, and of ARA at the end of the massage period, expressed as % of total fatty acids, were slightly significantly higher in the no oil group but with no difference when expressed as mg/mL (Table 4).

In an interesting though surprising way, the proportion of nervonic acid (C24:1n-9) and of C18:1n-9 DMA species were significantly higher in the phospholipids of the RBC membrane in neonates belonging to the no oil group (Figure 2A,B, respectively). Considering only the *p*-value obtained after adjustment on confounders (*p**), nervonic acid was significantly higher at discharge (+25%), while the C18:1n-9 DMA became higher immediately post-intervention (+52%) and remained higher at follow-up (+33.5%). When calculating the ratio [C18:1n-9 DMA/C18:1n-9 ] × 100, indicative of the relative amount of plasmalogen species containing C18:1n-9 at sn-1, values that were not different at the start of massage (5.91 ± 0.57% in no oil group vs. 5.28 ± 0.61% in the oil group, adjusted-*p* = 0.105) became significantly higher in the no oil group after adjustment for confounders (7.0 ± 3.0% vs. 4.6 ± 1.3% at end of massage, adjusted-*p* = 0.007; 8.4 ± 2.7% vs. 6.5 ± 1.6% at discharge, adjusted-*p* = 0.015; for the no oil vs. the oil group, respectively).

## 4. Discussion

Our study showed that a massage provided to neonates of 33–36 corrected gestational age, for five consecutive days, 10 min twice a day with 10 mL/kg/day isio4 oil, did not result in a higher growth or quicker digestive autonomy, in a lower hospital stay, or in changes in lipid metabolism compared to the same massage technique with no oil.

Concerning growth, a meta-analysis of eight studies concluded that massage with oil allowed for higher weight, higher absolute weight gain and higher weight gain velocity in preterm neonates at the end of the massage intervention, and a higher weight at 1–2-month follow-up [7]. In our study, such features were not reached. Indeed, even the average weight gain velocity at the end of massage period was 25.5% higher numerically in the oil group (average +2.5 g/kg/day over 5 days), the difference failed reaching significance. However, the conclusion raised in this meta-analysis was confusing since the authors compared data from the oil massage group with “a control group” that was either no oil massage or untreated control groups (no massage at all) without distinguishing the latter [7]. When exclusively considering data from oil massage neonates versus no oil massage, the results were more consistent with our findings. Indeed, in the study by Arora et al., the absolute weight gain (366–290 g) and the weight gain velocity (10.9 vs. 8.7 g/kg/day; average +2.2 g/kg/day) calculated over 28 days of massage (10 min 4 times a day) were also not significant, although numerically higher, in the oil massage (10 mL sunflower oil/kg/day; *n* = 20) versus the only massage group (*n* = 19), respectively [9]. Fallah et al. reported a higher weight gain but only at one month (*p* = 0.04) and at two months (*p* = 0.005) in premature neonates of about 35 weeks of gestational age being massaged with sunflower oil (*n* = 27) for 14 days, meaning that there may be benefits of a short-course of massage with oil versus no oil (*n* = 27) on weight gain that would be visible later in postnatal life [21]. Vaivre-Douret et al. showed no significant higher absolute weight gain after 21 days in 31–34 weeks of gestational age neonates (*n* = 12 per group) receiving for 10 days a massage (15 min session twice a day) with no oil (230 g), isio4 oil (301 g), or almond oil (216 g) [11]. In contrast to our study, Sankaranarayanan et al. reported a significantly higher weight gain velocity (+2.5 g/kg/day, *p* = 0.02) in about 35 weeks gestational age neonates after 31 days of massage (5 min, 4 times a day) when using coconut oil (11 g/kg/day, *n* = 38) versus no oil (8.5 g/kg/day, *n* = in 37) for massage [16]. In addition, Saeidi et al. studied neonates receiving MCT oil massage vs. massage only (5-min massage, 4 times a day for 7 days; *n* = 40 per group) and showed a higher weight gain in the oil group [22]. Comparing these few studies with our own, it seems likely that the discordance in conclusions was mainly due to the number of neonates studied, since the difference in weight gain velocity was close (about 11 g/kg/ day in oil groups vs. 8.5 g/kg/day in no oil groups) [9,16] and only significant for a headcount of at least 37 (lack of statistical power together with high variability between neonates). Of note, the weight gain velocity rates in our study as well as in others [9,16] were low, according to Fenton et al., who indicated that the optimal growth rates should range from 15 to 20 g/kg/day for infants born at 23–36 weeks [23]. In our study, 7 infants out of 18 reached a weight gain velocity of at least 15–20 g/kg/day in the oil group versus only 1 infant out of 18 in the no oil group. No superior increase in head circumference or in length after massage with isio4 versus massage alone in our study was in accordance with others who used sunflower or coconut oils [9,16].

With regard to mechanisms that might explain the weight improvement after combining massage and oil versus only massage, some authors have mentioned a possible absorption of oil lipids by the skin through a transpercutaneous absorptive route, leading to a higher supply of calories and nutrients [8,9,16,21,24]. The authors based this statement on an increase in triglyceride levels in neonate blood [8,24] and/or on changes in the fatty acid profile of blood lipids in accordance with the fatty acid composition of the oils used [8]. This proposal of a percutaneous absorptive route of lipids was first derived from studies reporting that topically applied drugs in premature newborns had a systemic delivery [25]. Then, some studies have reported in few neonates or adults the correction of a severe EFA deficiency (trienoic/tetraenoic ratio >0.4; low levels in LA and ARA in RBC or plasma) due to a long-term parenteral nutrition not supplied in lipids through the cutaneous application of sunflower or safflower oils [26,27]. However, there was no evidence of transcutaneous administration of EFA (especially LA) using safflower oil or safflower oil esters (at the dose of 1 g LA/kg/day) in parenterally fed premature neonates of 26–32 weeks of GA [28]. The use of oenethera oil providing cutaneous 1.9 g LA/kg/day for 20 days did not prevent the EFA deficiency induced by total parenteral nutrition in infants (increase in trienoic acid, decrease in LA and ARA) [29]. No increase in triglyceride concentration in plasma nor an increase in LA, ALA, and consequently in ARA or DHA of the plasma total lipids was observed in our study after using isio4 oil in massage. No change in plasma triglyceride concentration was also reported by another study using sunflower oil [9] and no change in plasma lipid fatty acid profile was reported in studies using soybean oil [30], safflower oil [28], or almond or isio4 oils [11]. To explain the discrepancy between studies using cutaneous vegetable oil application showing a raise in plasma triglycerides and/or fatty acid profile change [8,24] versus not [9,11,28,30], it was proposed that the oil lipids absorbed by the percutaneous route could have been stored in the subcutaneous adipose tissue without further systemic release [9]. In our opinion, the evidence of the existence of oil triglyceride absorption by the skin barrier delivering such molecules to subcutaneous adipose tissue or even to the bloodstream is still to come. Indeed, in our study, the amount of LA provided by oil massage during the 5-day intervention was higher (about 2.4 g/kg/day) than the one supplied by enteral feeding (about 0.35 g/100 mL human milk), suggesting that a rise in LA in bloodstream should have been expected but did not happen. In addition, by analogy with the mechanism of absorption of triglycerides through the intestinal barrier, it is well-known that the big molecules of triglyceride cannot pass through the intestinal epithelium before being split into smaller molecules (free fatty acids and 2-monyglycerides) by the action of digestive lipases [31]; this implies that a similar enzymatic phenomenon takes place at the skin barrier. Nevertheless, it cannot be ruled out that lipases from the skin resident microbiota can break oil triglycerides into free fatty acids [32]. However, it is more likely that oils applied on the skin provide lipids to keratinocytes, which metabolized them to build a functional epidermal barrier and restore the intracellular lipid in skin [10]. Of note, the neonates from our study were not likely to be EFA deficient. Indeed, the plasma trienoic acid/tetraenoic acid ratio was less than 0.4 (data not shown), and the EFA status index in RBC (averages from 1.6 to 1.8 all through the study period; data not shown) was relatively close to the values (i.e., 2.1) reported in 26–36 weeks healthy preterm newborns [33]. In addition, the LA concentration of total plasma lipids was in the upper range of values previously published in healthy premature neonates of 26–36 weeks of GA, themselves very similar to values found in healthy full-term newborns (i.e., 90 to 500 mg/L) [34]. The same remark can be made for the ARA concentration of plasma total lipids, which was within the range previously reported (120–270 mg/L) [34]. In the study by Soriano et al. [30] that showed no change in plasma LA and ARA levels in premature neonates after massage with soybean, neonates were also not EFA deficient (16–18% LA and 9.4–10% ARA of total FA in total plasma lipids).

Massage per itself can also improve neurodevelopment [4]. At our knowledge, only one study reported a superiority of oil massage versus only massage on neurological maturity score (based on oculomotor system, passive and active muscle tone of limbs, primitive reflexes) and on neuro-psychomotricity using several tests after 21 days [11] while other studies did not [9,16]. More recently, a long-term longitudinal study in 585 newborns reported a significant higher mental developmental quotient after massage with coconut oil versus massage only but at 6, 12, 18, and 24 months [35], indicating that benefits on neurodevelopment could be visible later after massage intervention, corresponding to a possible programming process. Our study did not allow for the observation of neurodevelopment through tools and scores usually used to explore such domains [9,11,16]. However, since a relationship does exist between enteric neural system (ENS) and brain development during the early postnatal period [36], an indirect measurement of development of the ENS in our study (i.e., time to reach enteral digestive autonomy) could give an idea of such a process. In fact, enteral autonomy was reached more quickly in the neonates massaged with no oil (*p* = 0.027), a difference that no longer persisted after adjustment to birth parameters (weight, GA), corrected GA at massage intervention start, and bronchopulmonary dysplasia status. Along the same line, our study highlighted a surprising observation in the group of neonates receiving massage with no oil, which was a significant higher level in the RBC membrane phospholipids of two molecules that are linked to neurodevelopment (i.e., nervonic acid (C24:1n-9 ) and plasmalogen species with a C18:1n-9 fatty alcohol bond at the sn-1 position of the glycerol backbone. Both differences persisted after adjustment to the confounding factors, suggesting that such factors were not explicative of the differences. The levels of RBC nervonic acid (3.0 ± 0.6% and 3.6 ± 0.8% of total FA, in oil and no oil groups, respectively) at the start of our study for preterm neonates of 32–37 weeks corrected GA were close to values previously reported (3.4 ± 1.7%) in Brazilian premature newborn of 26–36 weeks of GA [33]. Nervonic acid incorporation in the fetal brain increases rapidly in the last trimester of pregnancy [37], especially for white matter development and the myelination process [38]. Nervonic acid level in plasma phospholipids as well as in RBC membrane phospholipids is considered as a biomarker of brain maturation [38,39]. Thus, the significant higher level of RBC nervonic acid at discharge in our neonates massaged with no oil could be an indicator of a slightly higher brain maturation. Plasmalogens from the ethanolamine family (i.e., plasmenylethanolamine) represent 10–15% of total phospholipids in plasma membrane, a higher level being found in the brain or myelin structure [40]. These molecules are also well represented in the RBC membrane, whose analysis can be considered as a biomarker for organ levels [41]. Plasmalogens are specific phospholipids with a fatty acid alcohol bond at the sn-1 position (vinyl-ether linkage) of the glycerol backbone, mainly C16:0, C18:0, C18:1, and mono- or polyunsaturated fatty acids (such as DHA and ARA) at the sn-2 position [12,41]. During a methylation process, fatty alcohols are released and transformed into dimethyl acetals (DMA), allowing for indirect evaluation of plasmalogens [17,19], an analytical process that was used in our study. The values of C18:1 DMA that we reported in the RBC membrane phospholipids of our neonates were in the range of values previously published in adult humans (0.5% to 1.6%) [41]. Plasmalogens with C18:1 fatty alcohol at the sn-1 position on the glycerol backbone seems to be an important specie since a shift in DMA pattern from C18:1 to C18:0 was reported in plasmenylethanolamine in Alzheimer’s disease [42]. In addition, the ENS is linked to the central nervous system by the parasympathetic connections of the vague nerve, which innervates the gastrointestinal tract. A precursor of plasmenylethanolamine was reported to increase plasmalogen levels in the membrane of brain neurons as well as in the neurons of the myenteric plexus [43]. We can thus hypothesize that neuroprotection, in the broadest sense, could be enhanced through a higher level of plasmenylethanolamine carrying a C18:1n-9 fatty alcohol at the sn-1 position (indicated by a higher C18:1n-9 DMA in RBC phospholipids) in premature neonates who received massage with no oil. The reasons why such molecules that are important for neurodevelopment were higher in the neonates from the only massage group are difficult to explain in terms of real benefit of using oil or not for massage therapy. In other hand, we cannot rule out that such difference in RBC nervonic acid and plasmalogen C18:1n-9 species levels could be linked to confounding factors that we did not consider or/and to differences in the dietary amounts supplied to the neonates since both molecules are present in human milk in very variable ranges [18,19,44].

Our study had some limitations. First, the small size of our sample, which considerably reduced the probability of highlighting significant differences, if any; second, the possibility that the massage intervention was started too late from a “specific window” in the prematurity period (a limitation that is not certain since weight gain was also recorded in full-term neonates receiving oil massage [16]); third, we did not consider other confounding factors such as detailed nutritional intake; fourth, a longer follow-up (i.e., until at least two months of age after reaching a term corrected GA) was missing, which should have been necessary to observe differences.

Our study strengths are as follows. First, the consideration of several confounding factors (birth parameters, age at start of the massage, bronchopulmonary dysplasia status) for comparisons between groups, and second, a complete analysis of the trajectory of plasma and RBC lipids in neonates, which has rarely been conducted in the literature [8,11].

## 5. Conclusions

The use of isio4 as a massage oil was well tolerated as previously reported [11], but may not be the best oil to use in premature neonates [10]. The expected effect on a higher increase in weight gain at the end of massage intervention and on the follow-up was not reached with this oil in comparison to massage with no oil. The evolution of the concentration of lipids in plasma and of their fatty acid profile did not corroborate the existence of a percutaneous absorption of oil massage lipids through the bloodstream, possibly related to a permeability of the epidermal barrier function in premature neonates as postulated by other authors. Data provided by our study concludes that there was no significant change in the growth and digestive autonomy between massage with oil and no oil massage, which may or may not change later developmental outcomes. Further studies are needed to determine the real health interest of using oil during massage therapy in preterm neonates.

## Figures and Tables

**Figure 1 children-09-00463-f001:**
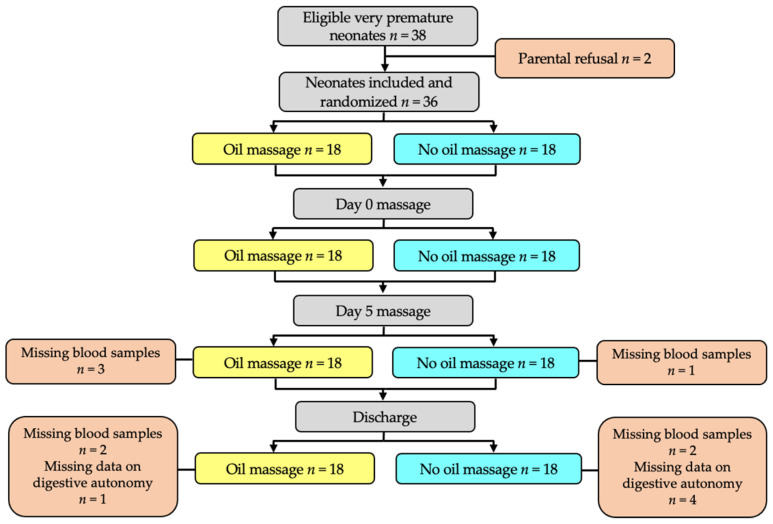
Flow chart of the study.

**Figure 2 children-09-00463-f002:**
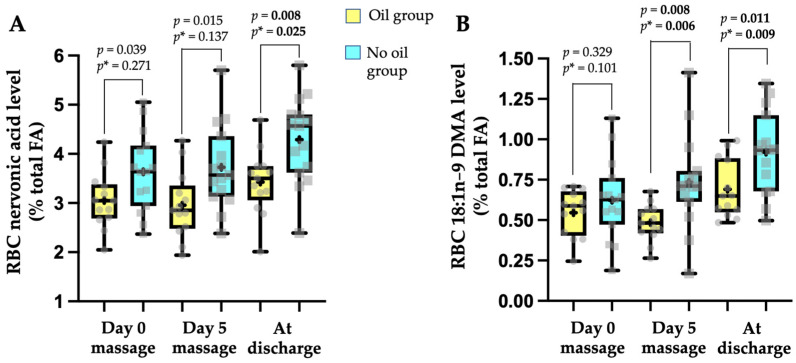
Box-and-whisker plot representation of (**A**) nervonic acid (C24:1n-9 ) or (**B**) C18:1n-9 dimethyl acetal levels in red blood cell membrane at the start (Day 0) and at the end (Day 5) of the massage intervention and at discharge in neonates receiving massage with oil (experimental group, *n* = 13, yellow) versus neonates receiving massage with no oil (control group, *n* = 15, blue). Data represented as plots are median, first quartile (Q1 or 25th percentile) and third quartile (Q3 or 75th percentile), min and max, mean (as a cross) and all individual values. Statistical analysis was performed using a general linear model for repeated measurement. *p* < 0.05 was considered as significant (indicated in bold). *p*-value not adjusted; *p**-value adjusted on birth weight, birth gestational age, corrected gestational age at day 0 of massage intervention, and bronchopulmonary dysplasia status (yes/no). Abbreviations: DMA, dimethyl acetal; FA, fatty acid; RBC, red blood cell.

**Table 1 children-09-00463-t001:** Studied population characteristics.

Variables ^1^	Oil Massage *n* = 18	No Oil Massage *n* = 18	*p*
**Antenatal**			
Corticosteroid treatment, *n* (%)	18 (100)	18 (100)	1.000
Premature rupture of membrane, *n* (%)	6 (33.3)	6 (33.3)	1.000
Premature labor, *n* (%)	6 (33.3)	8 (44.4)	0.494
Chorioamniotitis, *n* (%)	3 (16.7)	3 (16.7)	0.603
Vascular pathology, *n* (%)	8 (44.4)	8 (44.4)	1.000
Intrauterine growth restriction, *n* (%)	7 (38.9)	5 (27.8)	0.480
Fetal cardiac rhythm abnormality, *n* (%)	7 (38.9)	7 (38.9)	1.000
Singleton fetus, *n* (%)	11 (61.1)	12 (66.7)	1.000
Twin fetus, *n* (%)	4 (22.2)	3 (16.7)	1.000
Triples fetus, *n* (%)	3 (16.7)	3 (16.7)	1.000
**At birth**			
Gestational age (GA), weeks	28.9 ± 1.6	29.8 ± 1.1	**0.039**
Female gender, *n* (%)	11 (61.1)	10 (55.6)	0.735
Apgar at 5 min	8.6 ± 1.3	8.1 ± 1.6	0.415
Birth weight, g	1050 ± 24	1309 ± 288	**0.012**
Small for GA, *n* (%)	3 (16.7)	2 (11.1)	1.000
Head circumference, cm	25.4 ± 1.9	27.7 ± 1.6	**0.001**
Length, cm	35.3 ± 2.6	37.6 ± 2.3	**0.027**
**Postnatal**			
Exogenous surfactant therapy, *n* (%)	11 (61.1)	10 (55.6)	0.735
Respiratory distress syndrome, *n* (%)	17 (94.4)	17 (94.4)	1.000
Bronchopulmonary dysplasia 28 days, *n* (%)	10 (55.6)	4 (22.2)	**0.040**
Patent ductus arteriosus >7 days, *n* (%)	7 (38.9)	2 (11.1)	0.121
Necrotizing enterocolitis, *n* (%)	2 (11.1)	0 (0)	0.486
Number of nosocomial infections			0.126
None nosocomial infection, *n* (%)	7 (38.9)	12 (66.7)	
One nosocomial infection, *n* (%)	8 (44.4)	4 (22.2)	
Two nosocomial infections, *n* (%)	3 (16.7)	2 (11.1)	
**At inclusion day 0 massage**			
Corrected GA, weeks	35.2 ± 1.3	34.3 ± 1.0	**0.044**
Age, day	44 ± 16	31 ± 11	**0.018**
Weight, g	1725 ± 307	1867 ± 372	0.221

^1^ Data are given as mean ± SD or *n* (%). Statistical analyses were performed using Chi-2 or Fisher test for categorical variables, and Student *t*-test or Mann–Whitney for continuous variables, in accordance with the variables’ distribution; *p*-value of < 0.05 was considered significant. Abbreviations: GA, gestational age.

**Table 2 children-09-00463-t002:** Effects of massage on weight at immediate end of intervention and at discharge.

		Between Day 0 and after 5-Day Massage				Between the End of Massage and Discharge			
Variables ^1^	*n* Total	Oil Massage	No Oil Massage	Group Difference (95% IC)	*p*	Oil Massage	No Oil Massage	Group Difference (95% IC)	*p*
Absolute weight gain, g	36	128 ± 77	110 ± 46	−18.44 (−61.62;24.73)	0.391 *0.221	703 ± 343	659 ± 451	−43.61 (−315.10;227.87)	0.746 *0.739
Weight gain velocity, g/kg/day	36	12.3 ± 7.3	9.8 ± 4.1	−2.53 (−6.55;1.48)	0.209 ***0.108**	16.7 ± 7.3	16.9 ± 5.6	0.15 (−4.33;4.55)	0.945 *0.636

^1^ Data are given as mean ± SD. Statistical analysis was performed using a general linear model for repeated measurement; *p* < 0.05 considered significant. *p*-value not adjusted. * *p*-value after adjustment on birth weight, birth gestational age, corrected gestational age at day 0 of massage intervention, and bronchopulmonary dysplasia status (yes/no). *p* value close to 0.1 is in bold.

**Table 3 children-09-00463-t003:** Enteral autonomy, duration of hospital stay, and anthropometric parameters at discharge.

Variables ^1^	Oil Massage *n* = 18	No Oil Massage *n* = 18	*p*
Enteral autonomy corrected GA, weeks	37.5 ± 1.1 ^2^	36.9 ± 0.7 ^2^	0.126/*0.658
Enteral autonomy, day	61.5 ± 16.7	49.5 ± 10.2 ^3^	**0.027**/*0.626
Duration of hospital stay, day	23.6 ± 9.7	20.4 ± 15.0	0.136/*0.738
Corrected GA at discharge, weeks	39.2 ± 1.6	37.9 ± 2.5	**0.015**/*0.646
Age at discharge, day	72 ± 20	57 ± 18	**0.020**/*0.816
Head circumference at discharge, cm	33.1 ± 2.0	33.0 ± 2.0	0.987/*0.956
Length at discharge, cm	44.6 ± 2.7	44.9 ± 2.9	0.744/*0.808

^1^ Data are given as mean ± SD. ^2^ *n* = 17 for the oil group and *n* = 14 for the no oil group. ^3^ *n* = 15 for the no oil group. Statistical analysis was performed using the Student *t*-test or Mann–Whitney in accordance with the variables’ distribution; *p* < 0.05 was considered significant. *p*-value not adjusted. * *p*-value after adjustment on birth weight, birth gestational age, corrected gestational age at day 0 of massage intervention, and bronchopulmonary dysplasia status (yes/no). Abbreviation: GA, gestational age.

**Table 4 children-09-00463-t004:** Effects of massage on plasma lipid parameters and main polyunsaturated fatty acid composition.

		Day 0				Day 5				At Discharge			
Variables ^1^	Total *n*	Oil Massage	No Oil Massage	Group Difference (95% IC)	*p*	Oil Massage	No oil Massage	Group Difference (95% IC)	*p*	Oil Massage	No Oil Massage	Group Difference (95% IC)	*p*
**Lipid parameters g/L**													
Triglycerides	28	0.84 ± 0.43	0.69 ± 0.27	−0.15 (−0.43;0.12)	0.263/*0.284	0.93 ± 0.25	0.80 ± 0.39	−0.13 (−0.39;0.13)	0.321/*0.519	0.80 ± 0.19	0.70 ± 0.25	−0.10 (−0.28;0.07)	0.228/*0.464
Total cholesterol	28	1.11 ± 0.37	1.06 ± 0.26	−0.06 (−0.30;0.19)	0.644/*0.448	1.12 ± 0.26	1.07 ± 0.18	−0.04 (−0.22;0.13)	0.612/*0.439	1.02 ± 0.20	1.05 ± 0.18	0.03 (−0.12;0.18)	0.661/*0.575
**PUFA % total FA**													
** *PUFA n-6* **													
LA	28	16.03 ± 4.73	18.41 ± 4.24	2.38 (−1.11;5.86)	0.173/*0.210	18.37 ± 5.08	18.97 ± 4.95	0.60 (−3.30;4.51)	0.753/*0.991	20.09 ± 4.72	22.04 ± 2.78	1.95 (−1.01;4.91)	0.187/*0.488
ARA	28	4.90 ± 1.31	5.49 ± 0.69	0.59 (−0.21;1.38)	0.140/*0.193	4.84 ± 1.04	5.74 ± 1.18	0.90 (0.03;1.78)	**0.043**/*0.072	5.90 ± 1.69	6.24 ± 1.11	0.38 (−0,72;1.48)	0.485/*0.736
** *PUFA n-3* **													
ALA	28	0.29 ± 0.16	0.40 ± 0.17	0.11 (−0.02;0.24)	0.088/***0.039**	0.41 ± 0.23	0.40 ± 0.20	−0.006 (−0.17;0.16)	0.939/*0.642	0.52 ± 0.28	0.57 ± 0.21	0.05 (−0.15;0.24)	0.627/*0.929
DHA	28	1.17 ± 0.32	1.33 ± 0.23	0.16 (−0.05;0.37)	0.134/***0.044**	1.21 ± 0.38	1.37 ± 0.34	0.1 (−0.12;0.44)	0.248/*0.103	1.64 ± 0.59	1.82 ± 0.44	0.17 (−0.23;0.58)	0.381/*0.102
**PUFA g/L**													
** *PUFA n-6* **													
LA	28	0.549 ± 0.227	0.575 ± 0.187	0.03 (−0.14;0.19)	0.747/*0.753	0.676 ± 0.294	0.634 ± 0.238	−0.04 (−0.25;0.16)	0.680/*0.526	0.646 ± 0.184	0.671 ± 0.151	0.03 (−0.10;0.16)	0.686/*0.980
ARA	28	0.164 ± 0.050	0.175 ± 0.058	0.01 (−0.03;0.05)	0.604/*0.581	0.172 ± 0.051	0.191 ± 0.053	0.02 (−0.02;0.06)	0.341/*0.488	0.184 ± 0.043	0.190 ± 0.045	0.01 (−0.03;0.04)	0.731/*0.892
** *PUFA n-3* **													
ALA	28	0.010 ± 0.006	0.012 ± 0.006	0.002 (−0.002;0.01)	0.310/*0.108	0.016 ± 0.012	0.013 ± 0.009	−0.002 (−0.01;0.01)	0.603/*0.886	0.017 ± 0.011	0.018 ± 0.009	0.001 (−0.007;0.01)	0.886/*0.896
DHA	28	0.039 ± 0.011	0.041 ± 0.010	0.002 (−0.006;0.01)	0.588/*0.249	0.043 ± 0.013	0.046 ± 0.016	0.003 (−0.008;0.01)	0.570/*0.402	0.053 ± 0.012	0.056 ± 0.018	0.003 (−0.009;0.01)	0.640/*0.298

^1^ Data are given as mean ± SD. Statistical analysis was performed using a general linear model for repeated measurement; *p* < 0.05 was considered significant. *p*-value not adjusted. * *p*-value after adjustment on birth weight, birth gestational age, corrected gestational age at day 0 of massage intervention, and bronchopulmonary dysplasia status (yes/no). Abbreviations: ALA, alpha-linolenic acid; ARA, arachidonic acid; DHA, docosahexaenoic acid; FA, fatty acid; LA, linoleic acid; PUFA, polyunsaturated fatty acids.

## Data Availability

The datasets generated and/or analyzed during the current study are not publicly available since data belongs to the Assistance Publique des Hôpitaux de Marseille (AP-HM). However, datasets are available from the sponsor (promotion.interne@ap-hm.fr) on reasonable request and after signing a contract pertaining to the provision of data/or results.

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
