# Peer review of "Effect of Massage with Oil Balanced in Essential Fatty Acids on Development and Lipid Parameters in Very Premature Neonates: A Randomized, Controlled Study"

_children, 2022, doi:10.3390/children9040463_

Round 1

Reviewer 1 Report

Dear Authors

Thank you for giving us the opportunity to review your work. I am recommending following changes 

  • Good thorough spell check and basic grammer revision e.g. Line 102,103,104, 431
  • In abstract and introduction part, Primary aim should be revised as it states to measure neonatal development. Does that mean neuro-development  outcomes or author meant to say measurements like weight growth velocity etc. Because if the aim is to measure Neuro-develpomental outcomes, this study lacks the outpatient follow up data to look at infant development and if the primary aim was to measure weight , and other parameters as proxy for neuro-developmental outcomes, Authors need to state this in the aim. 
  • Same goes for conclusion, Its hard to make conclusion for developmental outcomes from the data provided. Data provided concludes that there was no changes in the parameters provided between two group which MAY  or may not change later developmental outcomes. 
  • Recommend Authors to really limit the discussion part and try to avoid unnecessary details 

Author Response

We are very grateful to the two reviewers for their time spent on our manuscript and for valuable comments which helped us improving our manuscript.

The changes made are highlighted in blue accordingly to reviewer #1’s or in green accordingly to reviewer #2 in the revised version of our manuscript. Changes highlighted in yellow are own corrections of mistakes found after re-reading again our manuscript (sorry about that).

We hope changes will be found satisfactory.  Thank you.

Reviewer #1

Thank you for giving us the opportunity to review your work. I am recommending following changes 

Question 1: Good thorough spell check and basic grammer revision e.g. Line 102,103,104, 431

Reply: Grammar and spelling have been checked using a specific function of the word software processor.

Question 2:  In abstract and introduction part, Primary aim should be revised as it states to measure neonatal development. Does that mean neuro-development outcomes or author meant to say measurements like weight growth velocity etc. Because if the aim is to measure Neuro-developmental outcomes, this study lacks the outpatient follow up data to look at infant development and if the primary aim was to measure weight, and other parameters as proxy for neuro-developmental outcomes, Authors need to state this in the aim. 

Reply: We agree this point is confusing as written. The primary objective was to measure neonatal development through the weight (weight in Kg, absolute weight gain in g and weight gain velocity in g/Kg/d), and via digestive autonomy as proxy for neuro-developmental outcomes. This point was clarified in the abstract and in the introduction part.

Question 3: Same goes for conclusion, Its hard to make conclusion for developmental outcomes from the data provided. Data provided concludes that there was no changes in the parameters provided between two group which MAY or may not change later developmental outcomes. 

Reply: We agree that “Data provided by our study concludes that there was no significant change in the growth and digestive autonomy between massage with oil and no oil massage which may or may not change later developmental outcomes”. This specific sentence was added in the conclusion part.

Question 4: Recommend Authors to really limit the discussion part and try to avoid unnecessary details 

Reply: we have tried limiting the discussion part as requested.

Reviewer 2 Report

This is a small RCT that assesses the effectiveness of massage with oil in preterm infants. RCTs including over 700 patients have been reported so it is unclear why this trial as designed to be rather small and if it was adequately powered. The results do not support that of the meta-analyses on the subject but it is possible that the group differences (control vs intervention) coupled with the small sample size led to a lack of difference in outcomes between the current trial and the meta-analysis.  

Abstract

What is the time frame for “immediately” after massage?

Introduction

The review of the gestational ages of various groups of preterm infants and the rest of the first paragraph are not pertinent and should be deleted.

Methods

Was a sample size estimate done? If you, provide the information.

It appears that some of the exclusion criteria were designed to occur after the intervention started. Is this correct? Did any baby get excluded like after randomization? It appears that all 36 patients randomized completed the trial.

A five day intervention was used. Is this sufficient to produce changes in weight if there would be a change? Per the clinical trial registration, the intervention was planned to be 10 days. Was this changed?

Results

Weight gain was poor in both groups. It is important to known the caloric intake per k that was given to the patients.

Minor

There are some opportunities to improve the English

It is unclear what amount of blood was taken each time or total (2 mL, 1mL/Kg?).

Author Response

We are very grateful to the two reviewers for their time spent on our manuscript and for valuable comments which helped us improving our manuscript.

The changes made are highlighted in blue accordingly to reviewer #1’s or in green accordingly to reviewer #2 in the revised version of our manuscript. Changes highlighted in yellow are own corrections of mistakes found after re-reading again our manuscript (sorry about that).

We hope changes will be found satisfactory.  Thank you.

Reviewer # 2

This is a small RCT that assesses the effectiveness of massage with oil in preterm infants. RCTs including over 700 patients have been reported so it is unclear why this trial as designed to be rather small and if it was adequately powered. The results do not support that of the meta-analyses on the subject but it is possible that the group differences (control vs intervention) coupled with the small sample size led to a lack of difference in outcomes between the current trial and the meta-analysis.  

Reply: As regards to your comment about RCTs including 700 neonates, we assume it is about the paper from Badr LK et al. 2015 MCN Am J Matern Nurs* (688 neonates in intervention groups from 34 RCTs). In this article, mostly all studies referenced are about massage alone versus control (no massage at all), and very few studies listed are about massage with oil versus control (no massage at all), i.e., study form Arora 2005 or from Kumar 2013, which are “pooled” in the meta-analysis with RCTs studying massage alone vs no massage. The question which interested us was: does the use of oil during massage improve further the weight gain or not? So, we had to design a study with a group of neonates receiving massage alone and a group of neonates receiving a massage with oil. Few studies have addressed this question by comparing a group of neonates with massage only with a group of neonates with massage plus oil, and the answers are still controversial. The study by Arora published in 2005 followed 20 very low birth weight neonates with oil massage and 19 with massage only for 10 min 4 times a day for 28 days (and 23 as control = no massage), and did not report a significant difference in weight gain. Sankaranaranayanan (2005) studied 38 neonates with massage with coconut oil and 37 with massage with mineral oil (and 37 controls with no massage) 4 times a day for 31 days and reported a higher weight gain velocity in coconut oil massage group compared to the no oil and the control groups. Fallah in 2013 studied 17 low birth weight neonates with sunflower oil massage and 17 with only massage 3 times a day for 14 days and reported a higher mean weight but only later in life at 1 and 2 months with oil. Saeidi (2015) studied 40 neonates receiving MCT oil massage vs 40 neonates having massage only (and 41 controls) 5 min massage 4 times a day for 7 days and reported a higher weight gain in the oil group compared to the two others. The meta-analysis of Li 2016 is confusing since the authors considered for a same study the data from the oil massage group vs data from the control group, which is a group without massage, while data from a massage alone group was available. So, this author concluded to higher weight gain in the case of massage with oil, but it was mostly versus no massage (since he used data from control = no massage instead of data from massage alone).

We have tried to better explain this in the discussion part. The article from Badr LK et al. and the one from Saeidi R et al. were added in the references.

Question 1: Abstract

- What is the time frame for “immediately” after massage?

Reply: massage was delivered for 5 days (day 0 to day 5, consecutively), and the day “immediately” after massage is the day 6, so the day just after the end of massage. The term “at the end of the massage period” was used now.

Question 2: Introduction

- The review of the gestational ages of various groups of preterm infants and the rest of the first paragraph are not pertinent and should be deleted.

Reply: The sentences about gestational ages of various groups of preemies have been deleted as requested.

Question 3: Methods

- Was a sample size estimate done? If you, provide the information.

Reply: A sample size estimate was done based on the study of Sankaranarayanan K et al. 2005 in which a significant higher weight gain velocity of about 3g/Kg/d was reported in preterm infants massaged with coconut oil compared to preterm infants massaged with no oil, the weight gain velocity being one of the primary outcomes of our study. Thus, to detect a difference in weight gain velocity of about 3g/Kg/d with a SD of 2.7, with a power of 90%, and error alpha of 0.05 (two-sided), 18 infants were required for each study group. This information was added in the manuscript.

- It appears that some of the exclusion criteria were designed to occur after the intervention started. Is this correct? Did any baby get excluded like after randomization? It appears that all 36 patients randomized completed the trial.

Reply: Yes, it is correct. No baby gets excluded after randomization because such exclusion criteria, as already indicated in “2.3. Studied population” were applied only “during the massage period” and none was observed during this specific period. When such events appeared, it was after the massage period and no baby has to be excluded. Yes, 36 infants randomized completed the trial (with missing data or blood samples for some as indicated in our flowchart in Figure 1).

Exclusion criteria were: “1) neonates who developed a co-morbidity during the massage period implying an interruption of this practice (e.g., nosocomial infection, necrotizing enterocolitis, respiratory deterioration leading to stop feeding and/or to put them again on CPAP or more invasive therapy;”

- A five-day intervention was used. Is this sufficient to produce changes in weight if there would be a change? Per the clinical trial registration, the intervention was planned to be 10 days. Was this changed?

Reply: As indicated in the introduction part, massage increases weight gain from 2 to 3 sessions per day lasting 10 to 15 minutes for 5 to 30 consecutive days (review from Niemi AK, Children (Basel) 2017). Thus, a massage lasting 5 consecutive days was reported in previous studies to be sufficient to produce changes in weight. The indication of 10 consecutive days in the clinical trial registration is due to an administrative error when entering the information, error that was never corrected while we asked for it. For all infants the massage was administrated for 5 consecutive days as planed since the beginning in this study.

Question 4: Results

- Weight gain was poor in both groups. It is important to know the caloric intake per k that was given to the patients.

Reply: According to Fenton TR et al. J Peds 2018, the optimal growth rates range 15-20g/Kg/d for infants born at 23-36 weeks. So, we agree that weight gain velocity in our study (about 12±7 g/Kg/d in isio4 oil massage group vs 9.8±4 in no oil massage one; infants of 33-36 weeks of corrected gestational age) was lower based on these values. As indicated in our paper, weight gain velocity ranged +2.2 to +25.8 g/Kg/d in our oil group (7 infants out of 18 reached values of at least 15-20g/Kg/d) and -1.3 to +15.1 g/Kg/d in our no oil group (only 1 infant out of 18 reached values of at least 15-20g/Kg/d). Nevertheless, it is to note that weight gain velocity with same magnitude than ours was reported also in the few studies who aimed with the same objective, i.e., 11±4 g/Kg/d in sunflower oil massage group vs 8.7±5 in no oil massage group (Arora J et al. 2005; infants of about 34 weeks of gestational age), 11±2.6 g/Kg/d in coconut oil massage group vs 8.4±2.7 g/Kg/d in no oil massage one (Sankaranarayanan K et al. 2005; infants of about 35 weeks of gestational age). This important point was added in our manuscript. The paper from Fenton TR was added in the references.

The caloric intake (kcal/Kg/d) was quite the same for all infants since they were all mostly enterally fed human milk following a clinical routine procedure (volume of parenteral perfusion < 60 mL/Kg/d).

Question 5: Minor

- There are some opportunities to improve the English

Reply: English was checked using DeepL traduction website and world software English correction.

- It is unclear what amount of blood was taken each time or total (2 mL, 1mL/Kg?).

Reply: sorry for this imprecision. The volume of collected blood was about 1 ml/Kg for each time, so three blood collections i.e., at day 0 of massage, day 6 (after the 5-day massage) and at discharge. This was corrected.